# Efficient Part-level 3D Object Generation
# via Dual Volume Packing

**Jiaxiang Tang**[1,2*]   **Ruijie Lu**[1]   **Zhaoshuo Li**[2]   **Zekun Hao**[2]   **Xuan Li**[2]
**Fangyin Wei**[2]   **Shuran Song**[2,3]   **Gang Zeng**[1]   **Ming-Yu Liu**[2]   **Tsung-Yi Lin**[2]
[1]State Key Laboratory of General AI, Peking University
[2]NVIDIA Research, [3]Stanford University

## Abstract

Recent progress in 3D object generation has greatly improved both the quality and efficiency. However, most existing methods generate a single mesh with all parts fused together, which limits the ability to edit or manipulate individual parts. A key challenge is that different objects may have a varying number of parts. To address this, we propose a new end-to-end framework for part-level 3D object generation. Given a single input image, our method generates high-quality 3D objects with an arbitrary number of complete and semantically meaningful parts. We introduce a dual volume packing strategy that organizes all parts into two complementary volumes, allowing for the creation of complete and interleaved parts that assemble into the final object. Experiments show that our model achieves better quality, diversity, and generalization than previous image-based part-level generation methods. Our project page is at https://research.nvidia.com/labs/dir/partpacker/.

## 1   Introduction

Part-level 3D generation focuses on creating objects composed of multiple distinct and semantically meaningful parts, which is crucial for downstream editing and manipulation in applications such as game development and robotics. Although recent methods have greatly improved the quality of 3D object generation, they typically produce fused shapes without explicit part structures [55, 56, 59, 5, 22, 54, 57]. Generating complete and meaningful 3D parts remains a major challenge. It requires a deeper understanding of both the global layout and local part interactions within the object, which current methods still struggle to model effectively.

Several recent methods [4, 52, 51, 28] have explored part-level 3D generation by first segmenting the fused mesh into incomplete parts (usually surface patches), and then applying reconstruction or completion models to each part individually. While these pipelines have achieved promising results, they suffer from two fundamental limitations. First, they rely heavily on external segmentation priors, often derived from 2D models or pretrained networks. This introduces additional preprocessing steps and poses a risk of error propagation—any mistake in the segmentation stage can negatively impact the final generation quality. Second, these methods process each part sequentially, resulting in inefficiencies during inference. As the number of parts increases, inference time scales linearly, regardless of the individual complexity of each part. These limitations underscore two core challenges in part-level 3D generation: handling an unknown and variable number of parts, and mitigating the inefficiency of part-by-part sequential processing.

In this paper, we present a novel approach for directly generating part-level 3D objects without relying on any 2D or 3D segmentation priors. Our framework enables end-to-end generation of

---

*This work is done while interning with NVIDIA.

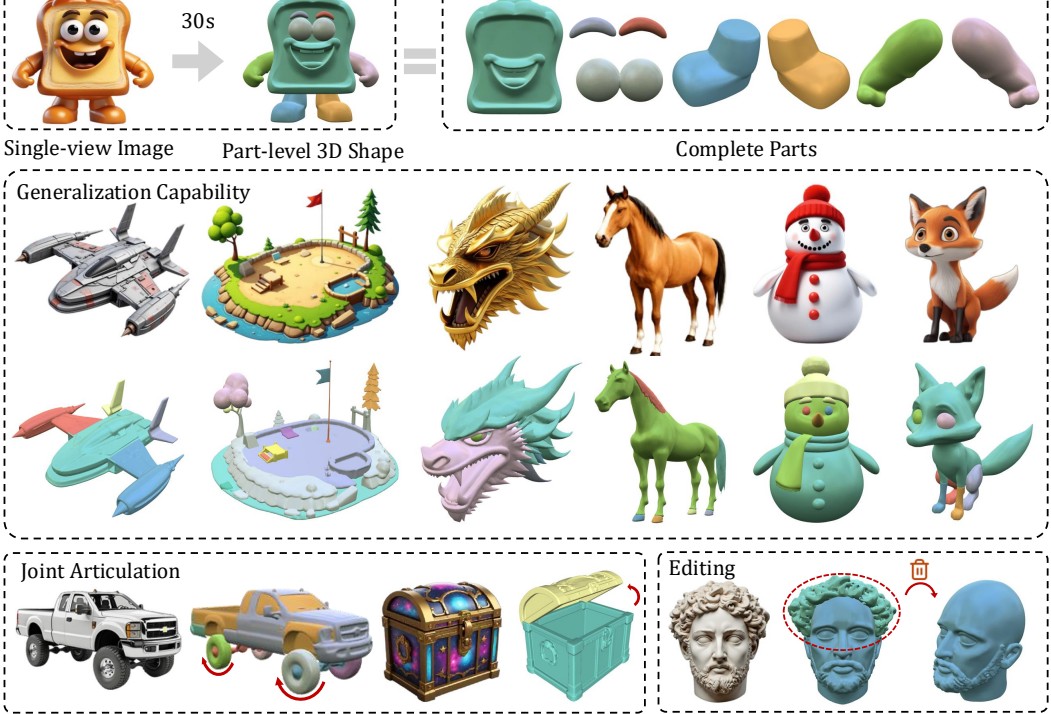

Figure 1: **End-to-end Part-level Image-to-3D Generation.** We present a method to generate 3D shape composed of individual and complete parts from a single-view image. Our method is trained only with 3D native information and can generate part-level meshes in about 30 seconds without relying on 2D segmentation prior models.

an arbitrary number of parts within a fixed time. We identify the key challenge to be the handling of overlapping regions between contacting parts. In contrast, disjoint parts, which are naturally separable as different connected components, can be processed in parallel rather than sequentially. This observation motivates our part-packing strategy, which aims to pack as many disjoint parts as possible into a shared volume to maximize space utilization, improve generation efficiency, and avoid fusing contacting parts. By analyzing the connectivity patterns commonly found in real-world objects, we observe that many can be effectively partitioned into two groups. We therefore formulate this task as a bipartite contraction problem and introduce a dual volume packing strategy, which fixes the output length and is fully compatible with existing 3D latent denoising models [56, 59].

In summary, our contributions are as follows:

1. We propose a novel end-to-end part-level 3D generation framework, which produces high-quality 3D shapes with an arbitrary number of semantically meaningful parts from a single input image, without relying on any segmentation prior.

2. We introduce a dual volume packing strategy that converts the part-connectivity graph into a bipartite graph through heuristic edge contraction, enabling efficient use of 3D volumes while preserving separation between contacting parts.

3. Experiments show that our method achieves more robust, diverse, and structurally consistent part-level generation compared to existing approaches, while offering a simpler and more efficient generation pipeline.

## 2 Related Work

### 2.1 3D Denoising Generative Models

3D-native denoising models for conditional 3D generation have seen substantial progress in recent years. Early research efforts focused on uncompressed 3D representations, such as point clouds [32], Neural Radiance Fields (NeRFs) [30, 17, 41], volumetric representations [13, 10, 33, 60, 31, 1, 2, 26, 50], and other formats [27, 3, 53, 47]. Despite their potential, these methods face limitations when applied to small or sparse datasets, often resulting in poor generalization and suboptimal quality.

More recent work has shifted towards latent denoising models for 3D generation [56, 57, 45, 21, 20, 16, 38, 9, 46, 22, 59]. These approaches typically employ a VAE to compress 3D data into a compact latent space, facilitating more efficient training of the denoising generative models. 3DShape2Vecset [55] first introduced a vector-set-based latent representation for effective 3D compression and generation. CLAY [56] demonstrates that combining VAE and 3D latent diffusion transformers yields strong performance on large-scale datasets and supports diverse conditioning modalities. TripoSG [22] pioneers the use of latent rectified flow [25, 24] and a mixture-of-experts architecture for 3D generation. Trellis [46] designs a sparse, structured latent representation with multi-format VAE decoders to support efficient rectified flow modeling. Hi3DGen [54] proposes a normal-bridge mechanism to enhance surface detail reconstruction. Dora [5] introduces salient edge sampling to improve the VAE's reconstruction fidelity, while Hunyuan3D-2 [59, 19] explores efficient network architectures to improve generation quality and accelerate VAE decoding. In this work, we extend such 3D latent denoising models to support part-level generation.

### 2.2 Part-level 3D Generation

While existing 3D latent denoising models can generate detailed geometry, their outputs are typically fused meshes extracted from occupancy or SDF fields, where all parts are fused together. However, many practical applications require part-level 3D meshes to support editing and manipulation. A core challenge in part-level 3D generation is the varying number of parts per object, whereas latent denoising models usually rely on a fixed-length latent code, making such variability difficult to handle.

One promising direction is to employ auto-regressive models, where next-token prediction naturally accommodates variable-length outputs. For instance, MeshGPT [37] introduces this concept by tokenizing a mesh using face sorting and VQ-VAE-based compression, followed by an auto-regressive transformer to predict the token sequence. Subsequent works [6, 43, 7, 8, 14, 40, 23, 58, 42, 44] expand on this idea by designing various mesh tokenization strategies and conditioning mechanisms, including point clouds and single-view images. However, these methods often face a major limitation: complex meshes typically contain a large number of faces, making the generation process computationally expensive and time-consuming.

Another line of work focuses on segmentation-then-completion frameworks that leverage 2D segmentation prior models [18, 35]. PartGen [4] adopts a multi-view diffusion model combined with 2D segmentation to generate multi-view part segmentation maps, which are subsequently used for multi-view completion and per-part 3D reconstruction. SAMPart3D [51] proposes a zero-shot 3D part segmentation approach that transfers part-level knowledge from 2D priors to the 3D domain, enabling multi-granularity segmentation. PartField [28] learns a feed-forward 3D feature field that encodes part semantics and hierarchical structure, allowing for clustering-based segmentation and shape-level correspondence. HoloPart [52] introduces a specialized 3D latent denoising model that completes each segmented part by attending to both local geometry and global context, thereby preserving structural consistency during reconstruction. Despite their effectiveness, these methods often rely on 2D segmentation priors, which may introduce propagation errors when the input segmentations are inaccurate or inconsistent. Moreover, their pipelines tend to be complex and inefficient, typically requiring iterative processing over individual parts. To overcome these limitations, we propose an end-to-end image-to-3D generation framework that directly produces part-level 3D meshes.

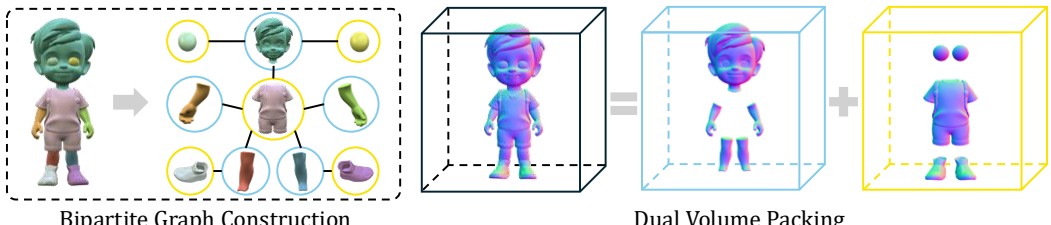

Bipartite Graph Construction                    Dual Volume Packing

Figure 2: **Dual Volume Packing.** Given a 3D mesh with part-level annotations, we propose to convert the part-connectivity graph into a bipartite graph, such that all parts can be packed into two volumes. Within each volume, parts do not contact each other, thus can be separated during mesh extraction.

## 3 Methodology

Similar to previous 3D latent denoising models [56, 22, 59, 54, 46], our model takes a single-view image as input and generates the corresponding 3D shape. In contrast, our goal is to predict separable and complete 3D parts that can be assembled into the final shape, rather than producing a single fused mesh. We begin by introducing our dual volume packing strategy (Section 3.1) and describing the process for curating a part-level dataset (Section 3.2). We then detail the architecture of our part-level 3D generation model and the associated training procedures (Section 3.3).

### 3.1 Dual Volume Packing

Most current 3D generation methods rely on volumetric representations, such as SDF grids, to represent shapes. Given an input image, these methods generate a single 3D volume, from which meshes can be extracted using iso-surfacing algorithms [29]. We observe that the key difficulty in part-level 3D generation lies in the handling of contacts between parts. If two parts are not in contact within the 3D space, they naturally form separate connected components in the extracted mesh and can be easily isolated. However, when parts are in contact, they become fused in the SDF grid, causing the loss of individual part information and making separation infeasible.

This observation motivates our concept of *volume packing*. **Instead of assigning each part to a separate volume, we propose packing as many disjoint parts as possible into the same volume** to maximize spatial efficiency and minimize computational cost. This problem can be formulated as a standard graph vertex coloring task. We construct a part-connectivity graph $\mathcal{G} = \mathcal{V}, \mathcal{E}$, where each 3D part corresponds to a vertex in $\mathcal{V}$, and each pair of contacting parts forms an edge in $\mathcal{E}$. The goal is to assign colors to vertices such that no two adjacent vertices share the same color, while minimizing the total number of colors. Parts with the same color can then be safely packed into a shared volume.

However, this strategy alone does not fully address the issue of varying output lengths. The upper bound of the chromatic number $\chi(G)$ depends on factors such as the graph's maximum degree and the presence of odd cycles. As illustrated in Figure 2, we further observe that many 3D objects have simple part-connectivity graphs with a bipartite structure, where $\chi(G) = 2$. This insight leads us to adopt a dual volume packing scheme, which is especially suitable for efficient part-level 3D latent denoising models. The output length becomes fixed, as we only need to generate two volumes, and there are only two valid colorings for each connected bipartite graph. Although it is possible to use more than two volumes, doing so introduces practical challenges: (1) When $\chi(G) \geq 3$, the graph often has multiple valid colorings, and the permutation invariance of color assignments complicates the learning process. (2) Using additional volumes results in lower space utilization per volume, which reduces overall efficiency. For these reasons, we adopt the dual volume packing strategy when curating our dataset.

**Bipartite Graph Extraction.** For meshes whose part-connectivity graphs are not bipartite, we apply a sequence of edge contraction operations (*i.e.*, merging specific pairs of parts) to transform the graph into a bipartite one. However, identifying the optimal set of edges to contract is known to be NP-hard [15]. We propose to use a heuristic algorithm specifically designed for the part-connectivity structures observed in meshes. Since bipartite graphs must not contain odd-length cycles, our method performs contractions to eliminate such cycles. To build the input graph, we conduct collision detection and connect an edge $e$ between each pair of collided parts. We slightly dilate each part

according to the resolution of the SDF grid to ensure that adjacent parts with tangential contact also form edges. The penetration depth between parts is used as the edge weight $w(e)$, with the intuition that parts with more collision should be fused. We then apply a depth-first search (DFS) algorithm to identify all cycles in the graph. To eliminate odd-length cycles, we greedily iterate over each odd cycle and contract the edge with the greatest penetration depth, effectively merging two vertices and converting the cycle into an even-length one. However, since some edges may be shared across multiple cycles, contracting a single edge can affect other cycles, potentially introducing new odd-length cycles. Therefore, we repeat this process over all cycles multiple times until no odd-length cycles remain. The full algorithm is provided in the supplementary materials.

## 3.2 Part-level Data Curation

Our packing algorithm requires 3D meshes with part-level annotations. To prepare the dataset for training, we perform part extraction, repair, and filtering to curate part-level data.

**Part Extraction.** As the 3D meshes are stored in `GLB` format, we extract their scene graphs and use them as the primary source of part annotations. In particular, for animated meshes, the geometry nodes typically correspond to animatable parts, serving as meaningful part annotations. However, many meshes contain only a single geometry node, with all geometric components fused into one. In such cases, we fall back on using connected components as a proxy for part annotations. Nevertheless, a semantically meaningful part may consist of multiple connected components. This issue is especially prevalent when the mesh has undergone UV unwrapping, which introduces seams in the otherwise watertight surface to flatten the 3D geometry into 2D space, resulting in multiple connected components and boundary loops. To address this, we apply a series of empirical post-processing rules: (1) We identify boundary loop pairs that share identical vertices, suggesting they originate from the same watertight surface, and merge the corresponding connected components into a single part. (2) Very small connected components are merged into an adjacent contacted part. (3) Pairs of connected components with a high intersection-over-union (IoU) score are merged into one part. While these rules do not perfectly recover ground-truth part annotations, they are sufficient for our training purposes.

**Part Repair.** When creating 3D models, artists may omit faces in occluded regions, resulting in non-watertight meshes. This issue is particularly common when meshes are separated into parts, leading to many non-watertight segments. However, VecSet-based VAEs [55] assume watertight meshes in order to learn a balanced signed distance field (SDF). A common workaround is to dilate the surface into a thin shell surrounding the geometry [56, 5], creating a watertight shape. While this technique ensures watertightness, it is undesirable because it distorts the original geometry preferred by artists and introduces unbalanced distribution of SDF values. Instead, we propose stitching certain boundary loops exposed during part extraction. This operation reduces the number of non-watertight parts and improves the balance of the resulting SDF grid for training.

**Data Filtering.** Given the two volumes of parts, ideally the amount of occupied space within each volume are roughly the same for training stability. However, some data samples may still contain extremely unbalanced distribution after the heuristic part extraction and repair processes. To ensure dataset quality, we introduce a filtering step to remove the unbalanced examples. Let $o_1$ and $o_2$ denote the occupancy ratios of the two packed volumes. We discard samples that satisfy the following condition:

$$((o_1 < 0.001) \wedge (o_2 < 0.001)) \vee \left( \frac{\min(o_1, o_2)}{\max(o_1, o_2)} < 0.1 \right) \tag{1}$$

This criterion effectively filters out data with highly unbalanced SDF grids, which are difficult to learn and less useful for model training.

## 3.3 Dual latent Generation

Our model builds upon several designs from previous VecSet-based latent denoising models [55], notably incorporating the salient edge sampling strategy from Dora [5] and the rectified flow model from Trellis [46]. As illustrated in Figure 3, the model is composed of three main modules. A VAE encodes the packed volumes into a compact latent code, which can later be decoded into separable mesh parts. DINOv2 [34] is used as the image encoder to extract conditional features for

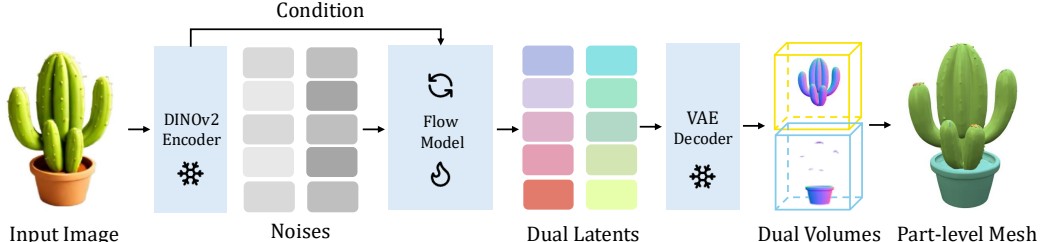

Figure 3: **Network Architecture.** Our model takes a single-view image as the input condition, and generate the dual latents at the same time with a flow model. The latents are decoded to dual volumes, which can be divided into parts and assembled back to the whole mesh.

cross-attention. Finally, a rectified flow model is trained to denoise the latent codes conditioned on these image features.

**VAE Model.** The VAE consists of a dual cross-attention encoder [5] and a self-attention decoder. Both uniform point samples and salient edge point samples are fed into the encoder. We primarily stack self-attention layers in the decoder and omit them in the encoder to accelerate the encoding process, which is invoked repeatedly during flow model training. The encoder's intermediate features are compressed into a latent code. Following previous works [22, 59, 5], we train the VAE with multiple latent code sizes to support progressive training of the flow model and to improve convergence speed.

**Flow Model.** The flow model consists of a stack of attention layers, following similar designs as in [22, 59, 56]. A key difference is that we denoise a pair of latent codes simultaneously, rather than a single one. The two latent codes are concatenated, allowing information exchange through the attention layers. To differentiate between them, we add a learnable part embedding exclusively to the second latent code as we find this helps to reduce duplicate parts. During inference, both latent codes are predicted jointly and decoded into two separate volumes, each containing disjoint mesh parts. These parts are then assembled to reconstruct the complete 3D shape.

## 4 Experiments

### 4.1 Implementation Details

**Dataset.** We use the Trellis500k subset [46] of the Objaverse-XL dataset [12, 11] (ODC-BY v1.0 license). After applying our part extraction process (Section 3.2), approximately 386K meshes with more than one part remain. Further filtering results in around 254K meshes with well-balanced SDF grids, which are used for training our model. Specifically, we pack the extracted parts into two separate volumes and follow the data preparation pipeline established in prior work [5]: (1) Each volume is converted into a watertight mesh [56], and are then used to extract uniform surface samples, salient edge samples, and point-SDF pairs to train the VAE [5]. All meshes are normalized to the $[-0.95, 0.95]^3$ cube, and watertight conversion is performed at a resolution of $512^3$, with the dilation threshold set to the voxel size. (2) Each mesh is rendered from multiple camera viewpoints to serve as the conditional input images. Following [46], we do not align the image poses with the geometry. Instead, the model is trained to learn pose-invariant generation by mapping diverse viewpoints to the canonical space.

**Training.** Our model is trained in multiple stages. We first pretrain the base VAE and flow model using the fused shape dataset without incorporating part-level information. The VAE is trained at multiple latent sizes on 64 A100 GPUs over the course of approximately one week. Next, the flow model is trained progressively with increasing latent sizes [56, 59, 5]. This progressive training phase spans about two weeks using at most 256 A100 GPUs. In the part-level stage, we observe that the VAE occasionally produces artifacts when reconstructing small parts. To mitigate this issue, we first finetune the VAE using part-level shapes. For flow model finetuning, we use the largest latent size, resulting in a total latent dimensionality of $4096 \times 2 = 8192$. All parameters are inherited from the pretrained model except for the newly introduced part embedding layer. The finetuning takes about two weeks using 256 A100 GPUs. Please refer to the supplementary materials for additional implementation details.

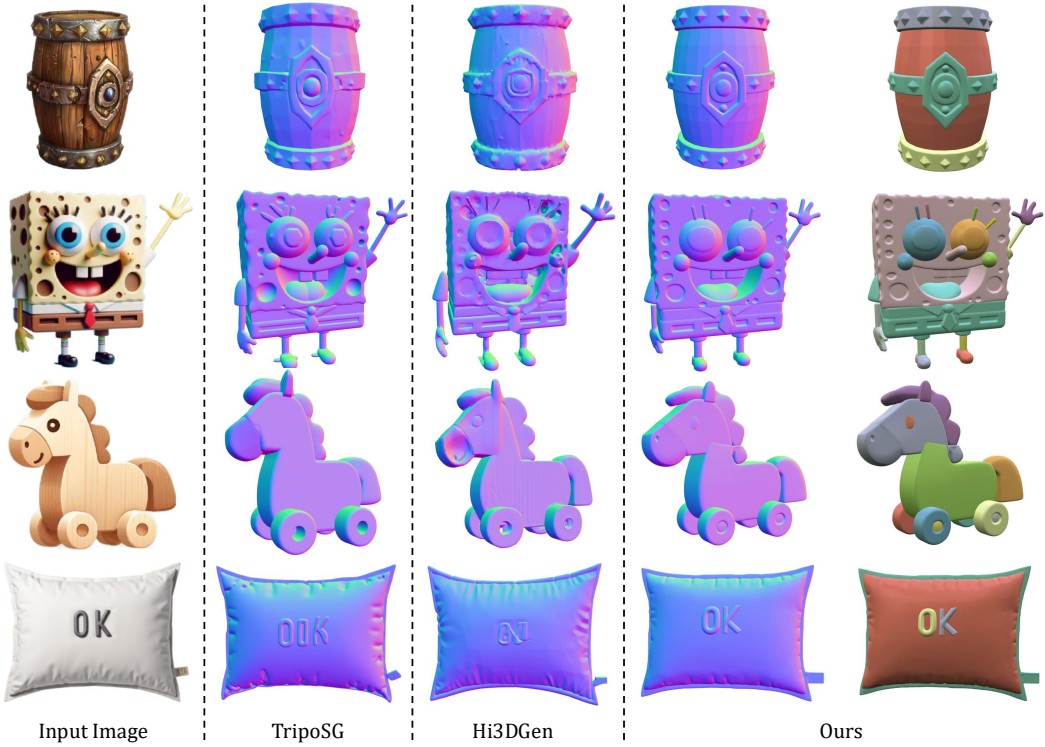

| Input Image | TripoSG | Hi3DGen | Ours |

Figure 4: **Comparison on Image-to-3D Generation.** Our method generates part-level meshes with competitive quality from single-view images compared to previous methods.

## 4.2 Qualitative Comparisons.

Our model is a latent denoising framework for image-to-3D generation, following the design principles of previous works [22, 46, 54, 59, 5]. We first evaluate the quality of 3D generation from single-view images. Figure 4 presents a qualitative comparison between our results and those generated by TripoSG [22] and Hi3DGen [54]. Our model achieves comparable or superior visual quality, demonstrating stronger alignment with the input image and producing more aesthetically pleasing mesh surfaces.

Unlike these baselines, which generate a fused shape in a single volume, our model can directly generate individual parts from a single-view image in an end-to-end manner. For part-level 3D generation, we primarily compare against HoloPart [52]. TripoSG [22] is used to generate the fused 3D mesh. We find that the original segmentation methods [51, 39] suggested by HoloPart are slow and error-prone, so we apply PartField [28] to obtain 3D segmentation masks. Following [28], we run with different cluster counts (number of parts) for each object and select the most reasonable one manually. As shown in Figure 5, our method produces more reasonable parts.

Furthermore, since our model is based on the rectified flow framework [25, 24], by varying the initial noise, we can generate diverse outputs from the same input. Figure 6 showcases examples of diverse generation results under different random seeds.

## 4.3 Quantitative Comparisons.

We also evaluate the generation quality of image-to-3D methods from single-view images. Following Hunyuan3D-2.0 [59], we adopt ULIP [48, 49] and Uni3D [61] as evaluation metrics. These models learn unified representations across text, image, and point cloud modalities, enabling cosine similarity-based comparisons between generated 3D shapes and reference images. To perform the evaluation, we curate a test set of 40 images sourced from diverse domains, and use each method to generate corresponding 3D meshes. To ensure a fair comparison, we fix the number of denoising steps to 50, extract meshes at a grid resolution of $512^3$, and simplify them to $50000$ faces via decimation. Since

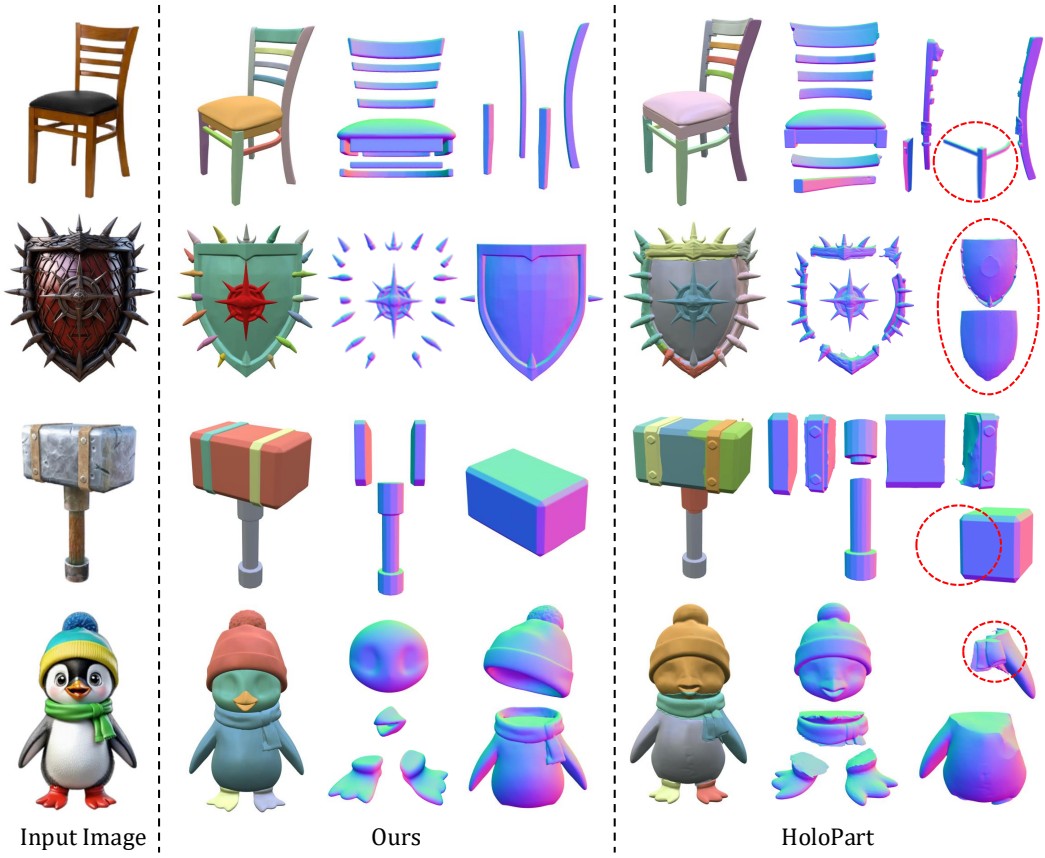

| Input Image | Ours | HoloPart |
|---|---|---|

Figure 5: **Comparison on Part-level 3D Generation.** Our method directly generate complete parts, while other methods require mesh segmentation and part completion.

| Method | Single Volume | | | | Dual Volumes |
|---|---|---|---|---|---|
| | Hunyuan3D-2 [59] | Hi3DGen [54] | TripoSG [22] | Ours† | Ours |
| ULIP [48] | 0.1609 | 0.1641 | 0.1726 | **0.1729** | 0.1715 |
| ULIP-2 [49] | 0.3962 | 0.3944 | **0.4048** | 0.4022 | 0.3986 |
| Uni3D [61] | 0.3872 | 0.3864 | 0.3912 | **0.3941** | 0.3906 |

Table 1: **Comparison of image-to-3D generation.** We measure the cosine similarity between generated meshes and input images using different feature extractors. † Our model pretrained on fused shapes to initialize our part-level model.

both ULIP-2 and Uni3D require colored point clouds as input, we assign a uniform white color to all mesh outputs before computing the metrics. As shown in Table 1, our model pretrained on fused shapes achieves competitive performance, aligning well with qualitative visual results. While the part-level fine-tuned model yields slightly lower scores due to its structural decomposition focus, it remains comparable to recent approaches, demonstrating a favorable trade-off between condition following and part-aware generation.

We further compare inference speed in the context of part-level generation. Specifically, HoloPart [52] requires a segmented mesh as input, which depends on both an image-to-3D model and a mesh segmentation model [39, 51, 28]. These two preprocessing steps alone take several minutes, and the subsequent part-wise completion time increases linearly with the number of parts. In contrast, our method takes a single-view image and directly outputs 3D parts in about 30 seconds no matter how many parts it contains, achieving huge acceleration.

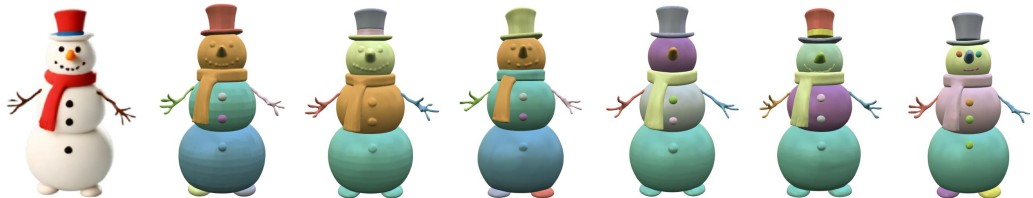

Figure 6: **Diversity on Image-to-3D Generation.** We can generate diverse and meaningful parts with different random seeds.

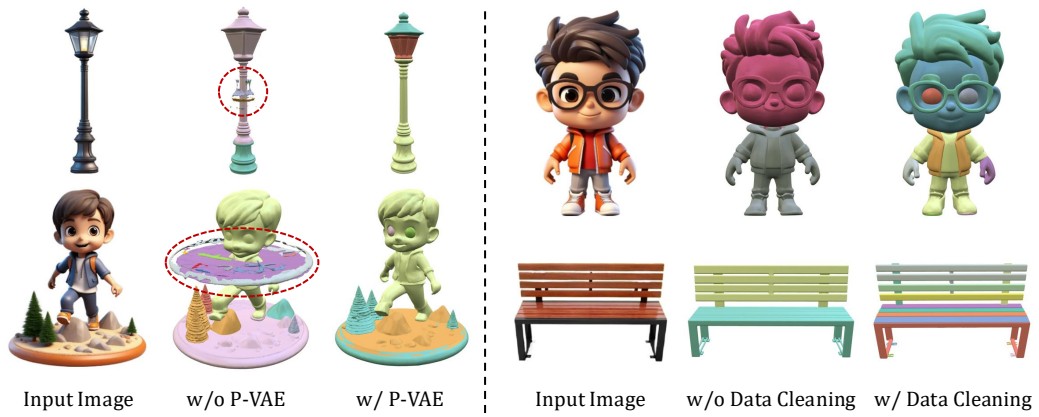

| Input Image | w/o P-VAE | w/ P-VAE | Input Image | w/o Data Cleaning | w/ Data Cleaning |

Figure 7: **Ablation Study.** We ablate different training designs and compare the generation quality.

## 4.4 Ablation Study

We conduct ablation studies on key design choices in the part-level finetuning stage, as shown in Figure 7. First, we evaluate the impact of part-level finetuning on the VAE (denoted as P-VAE). The VAE is originally pretrained on fused shapes that are normalized and recentered; when directly applied to part-level data, which may not be centered, it often introduces artifacts. After finetuning on part-level data, the VAE becomes more robust to such spatial variations. Second, we assess the effect of the data filtering process. Without filtering, the model tends to generate overly simplistic segmentations (often consisting of only two parts), whereas filtering the training data leads to improved segmentation quality and greater diversity. Lastly, we also validate the necessity of the part embedding. We find that without adding this learnable embedding to the second latent code, the model struggles to differentiate between the dual volumes and generates overlapping or identical parts.

## 4.5 Limitations and Future Work

Our method still exhibits several notable limitations: (1) Limited control over part granularity. The part extraction process relies solely on scene graph information from the input meshes, which are often noisy or inconsistent across the dataset. As a result, the extracted part divisions are diverse but unpredictable, hindering both consistency and user control. Incorporating additional input such as a segmentation mask may offer improved part-level controllability. (2) Constraints of bipartite contraction and dual-volume packing. While these strategies serve as practical solutions for managing complex part relationships, they are inherently limited in expressive capacity. For instance, three mutually contacting parts cannot be represented using only two volumes, leading to suboptimal results. A potential solution is to allow for more volumes, such as converting the connectivity graph into a planar graph and applying the four-color theorem [36] for packing. These limitations still constrain the model's capacity for precise editing and joint articulation of any 3D objects.

# 5 Conclusion

In this paper, we presented a novel end-to-end framework for part-level 3D generation from a single image. By leveraging 3D part connectivity and introducing a dual volume packing strategy, our method avoids reliance on 2D segmentation priors and efficiently handles an arbitrary number of parts. Experimental results demonstrate that our method achieves high-quality, diverse, and efficient part-level mesh generation, outperforming existing baselines. This work offers a promising step toward editable and structured 3D content creation for downstream applications.

**Acknowledgements.** This work is supported by the National Key Research and Development Program of China (2020YFB1708002), National Natural Science Foundation of China (61632003, 61375022, 61403005), Grant SCITLAB-20017 of Intelligent Terminal Key Laboratory of SiChuan Province, Beijing Advanced Innovation Center for Intelligent Robots and Systems (2018IRS11), and PEK-SenseTime Joint Laboratory of Machine Vision.

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

# A  More Implementation Details

## A.1  Bipartite Contraction

We detail the greedy odd-cycle contraction algorithm used in the dual volume packing process in Algorithm 1. The algorithm first performs a depth-first search to identify all cycles (both even and odd) in the graph. It then iteratively processes each odd cycle by contracting the edge with the largest weight, repeating this step until no odd cycles remain. In Figure 8, we present the distribution of the number of parts in our processed dataset. We observe that approximately 60% of the data samples contain fewer than 10 parts. Only 5% of the samples have more than 200 parts. Since enumerating all simple cycles in a graph is not a polynomial-time operation and can be time-consuming for graphs with dense edge structures, we restrict the use of this algorithm to graphs with fewer than 100 edges to ensure processing efficiency. For more complex graphs, we directly apply two-coloring and leave the conflicting edges where both vertices have the same color as contracted edges.

---

**Algorithm 1:** Greedy Odd Cycle Contraction

**Data:** Graph $\mathcal{G} = \{\mathcal{V}, \mathcal{E}\}$, where $\mathcal{V} = \{v_i\}_{i=1}^N$, $\mathcal{E} = \{e_i\}_{i=1}^M$.
**Result:** Array **O** to hold the edges to contract.

```
/* Find all cycles C = {Ci}, where each cycle Ci = {ej}          */
C = DFS (G);

/* Loop and contract until there are no odd cycles.             */
while C contains odd cycles:
    for C in C:
        if C is an odd cycle:
            /* Find the edge e with the largest weight in C.     */
            e = arg max_{e'∈C} w(e');
            /* Mark this edge for contraction.                   */
            O.append(e);
            /* Remove this edge from all cycles if exists.       */
            for C' in C:
                C'.remove(e);
return O;
```

---

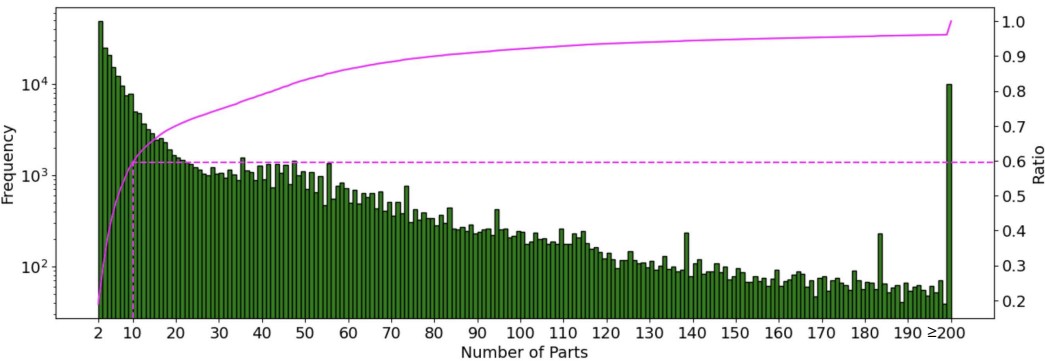

Figure 8: Statistics on the number of part in the processed dataset.

## A.2  Surface Sampling

Through dual volume packing, we split the raw mesh into two sub-meshes. For each sub-mesh, we compute the unsigned distance field (UDF) on a $512^3$ grid. To determine the empty region (positive distance), we apply a flood-filling algorithm starting from a corner voxel (e.g., the voxel at index $[0, 0, 0]$), which is guaranteed to be outside the mesh since all meshes are normalized to the range $[-0.95, 0.95]^3$. We then define the complement as the occupied region (negative distance), yielding a

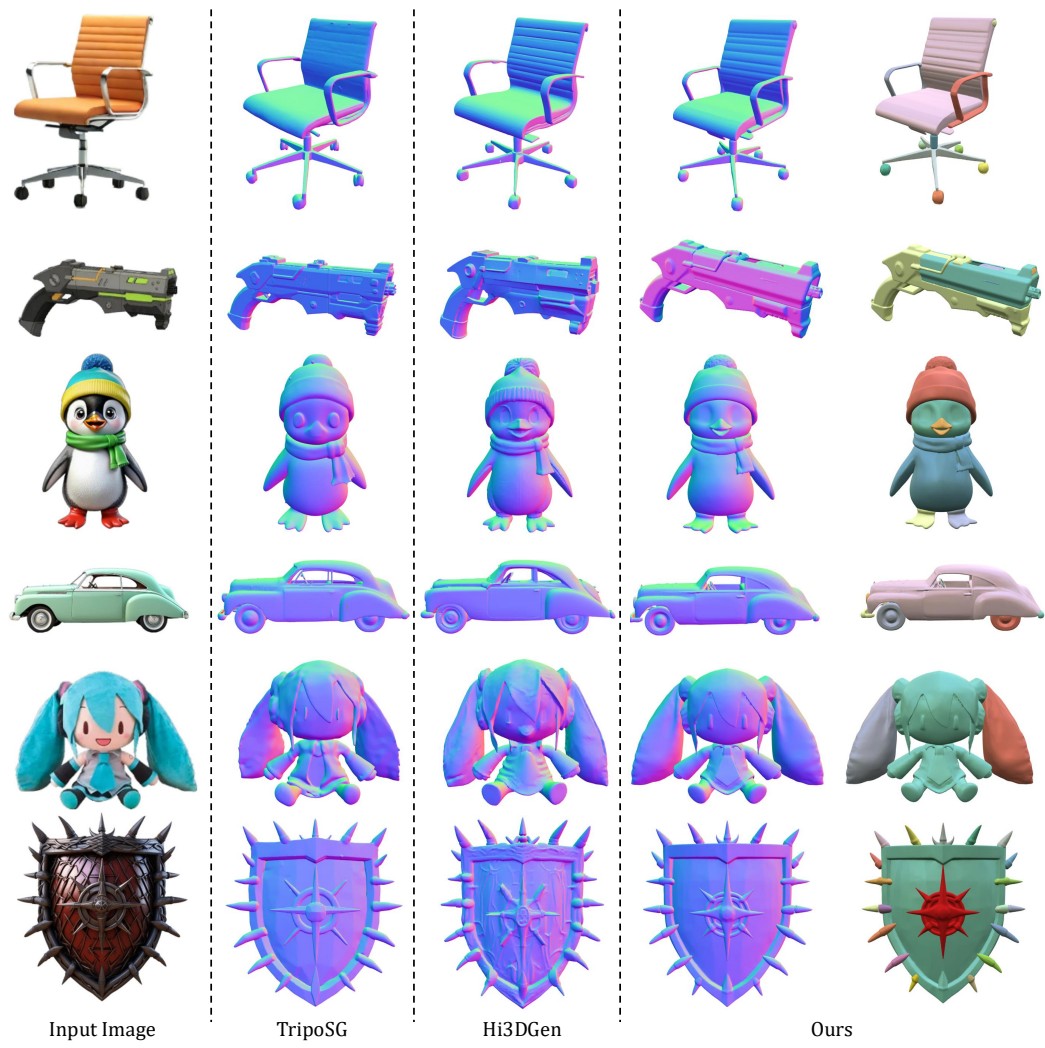

| Input Image | TripoSG | Hi3DGen | Ours |

Figure 9: **More Comparisons on Image-to-3D Generation.**

signed distance field. Using Marching Cubes [29], we extract a watertight surface from the resulting signed distance field.

Following Dora [5], we sample 32768 uniformly distributed surface points and 16384 salient edge points as input to the VAE. The dihedral angle threshold for salient edge detection is set to less than $165°$. If no salient edges are present in the mesh, we randomly subsample from the uniform surface points to serve as salient edge samples. To supervise the signed distance function (SDF) output, we further generate and store three types of point-SDF pairs: (1) Uniformly sampled points in the volume $[-1, 1]^3$; (2) Near-surface point samples; (3) Near-salient-edge point samples.

During VAE training, we randomly select 16384 uniform samples, 8192 near-surface samples, and 8192 near-salient-edge samples in each iteration.

# B  More Results

## B.1  Qualitative Comparisons

We provide more qualitative comparisons in Figure 9. Our method outperforms prior approaches in most cases, particularly in preserving fine-grained geometric details and producing complete

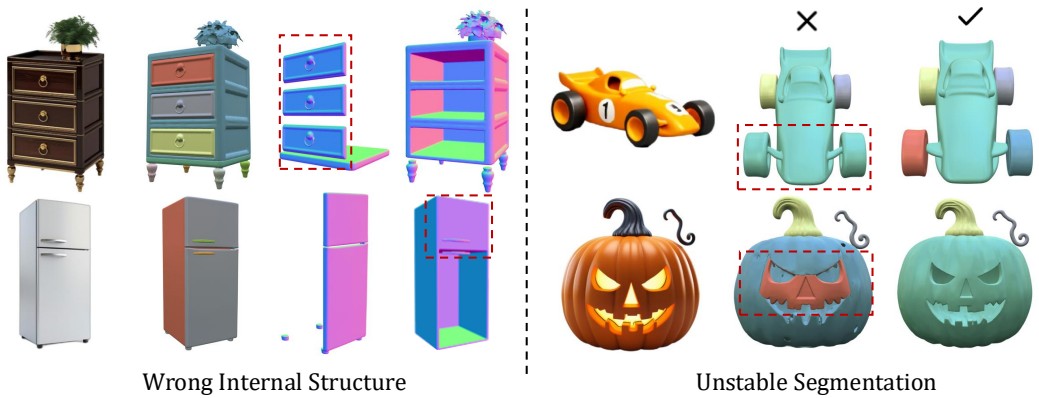

Wrong Internal Structure                    Unstable Segmentation

Figure 10: **Limitations and Failure Cases**. We showcase some failure cases of our model.

parts. Compared with baseline methods, our generated shapes exhibit better surface smoothness, part separation, and overall fidelity.

## B.2   Failure Cases

In Figure 10, we show some typical failure cases of our model. The first type involves incorrect internal structures. For shapes that contain intricate internal components, the underlying part connectivity graph becomes highly entangled. In such cases, our greedy odd cycle contraction algorithm may fail to partition the mesh into two meaningful groups, resulting in incomplete or incorrect reconstructions. For example, in some drawer-like objects, the inner compartment is not properly connected to the outer casing, leading to broken or missing internal parts. The second type of failure relates to unstable segmentation. Due to inconsistencies in part annotations within the training dataset and the absence of explicit part granularity control, our model can produce varying segmentation results when given different random seeds. While some samples yield satisfactory and semantically aligned part divisions, others may result in undesirable fusions or erroneous splits. For instance, the wheels of a car may sometimes be merged with the car body, violating the expected part separation. This instability highlights the need for more reliable part annotations and stronger priors to enforce consistent structural decomposition.

