# OpenReview forum: "Efficient Part-level 3D Object Generation via Dual Volume Packing"
_NeurIPS.cc/2025/Conference — NeurIPS 2025 poster_

### Official Review · Reviewer_Gc4w · 2025-06-24

**Clarity:** 3
**Significance:** 4
**Originality:** 3
**Rating:** 4
**Confidence:** 5

**Summary:**

This paper proposes a novel end-to-end framework for part-level 3D object generation from
a single image. Unlike previous approaches that generate fused meshes or require external
segmentation priors, the method directly predicts multiple complete and semantically
meaningful parts. The core contribution is a dual-volume packing strategy, which organizes
parts into two separate volumes based on their spatial connectivity. This bipartite-based
packing allows the model to efficiently handle arbitrary numbers of disjoint or contacting
parts while maintaining structural separation. The framework is trained using a large-scale
curated dataset derived from Objaverse-XL, with extensive preprocessing to ensure
part-level consistency and watertight geometry. The model leverages a VAE and a rectified
flow-based denoising module conditioned on image features extracted by DINOv2.
Experimental results demonstrate that the proposed method outperforms prior baselines in
visual quality and structural coherence, offering diverse, editable 3D outputs suitable for
downstream applications such as content creation or robotics.

**Questions:**

Given that the method generates 3D objects from a single input image, could the authors
clarify if and how the approach addresses the reconstruction or completion of unseen or
occluded parts of the object? Understanding this aspect is important to assess the model’s
capability for generating complete and realistic 3D shapes from limited views.

**Ethical Concerns:**

["NO or VERY MINOR ethics concerns only"]

**Final Justification:**

The author addressed my concerns well, so I will raise my judgment.

**Limitations:**

Since the number of parts is not provided as an input, the resulting segmentations may not
reflect the user’s desired level of granularity. This limits user control and reduces the
consistency and predictability of the output.

**Paper Formatting Concerns:**

No formatting concerns.

**Quality:**

2

**Strengths And Weaknesses:**

Strengths:
The proposed framework directly predicts complete and semantically meaningful 3D
parts from a single image, without relying on 2D/3D segmentation priors. This marks
a clear advancement over prior methods that depend on fused meshes or
segmentation-based pipelines.

The introduction of a dual volume packing strategy, where the part-connectivity graph
is transformed into a bipartite structure via heuristic edge contraction, is technically
novel and practically effective. It enables efficient packing of disjoint parts while
preserving structural separation, addressing a key challenge in part-level generation.

Although the training phase is resource-intensive, the inference process is efficient
and scales independently of the number of parts, an advantage over sequential
part-wise methods.

Weaknesses:
Compared to existing models, our method requires substantially more computational
resources. For example, TripoSG takes about 3 weeks of training on 160 A100
GPUs, and Hi3DGen uses only 8 NVIDIA A800 GPUs, whereas this paper’s
approach demands up to 256 A100 GPUs over multiple weeks. This high cost may
limit practical adoption and accessibility.

The ablation study presented in Figure 7 primarily shows qualitative comparisons
such as “/ w/o P-VAE / w/ P-VAE” and “w/o Data Cleaning / w/ Data Cleaning”
focusing on generation quality. However, the paper lacks quantitative evaluation
metrics for these design choices.

Additionally, even for the same object category (e.g., a tree), the model exhibits
inconsistent behavior depending on the scale: larger-scale instances tend to be
represented as a single part, while smaller-scale instances are segmented into
multiple parts such as individual leaves.

The paper includes joint articulation and editing as visual examples in Figure 1,
despite not implementing or evaluating these capabilities in the proposed model.
While these applications may serve as motivation, their inclusion without empirical
support or clarification may mislead readers into assuming that the method supports
such functionalities. This weakens the clarity and transparency of the paper’s scope
and contributions.

While not strictly necessary for this kind of paper, a user study or at least user-centric
evaluation would have strengthened the claims around usability  and editability,
especially since interactive applications are a key motivation.

---

> ### Author Rebuttal · Authors · 2025-07-28
>
> Thank you for your valuable time and insightful comments! We have tried to address your concerns in our rebuttal:
>
> **Q:  The method requires more computational resources.**
>
> We respectfully disagree. Our VectorSet-based 3D diffusion model’s computational requirements (256 A100 GPUs for 2 weeks) align closely with comparable models like CLAY (256 A800 GPUs for 15 days) and TripoSG (160 A100 GPUs for 3 weeks). Hi3DGen benefits from fine-tuning the efficient sparse volume latent representation of TRELLIS. Furthermore, since the model is progressively trained in multiple stages similar to CLAY, extended training yields diminishing returns and allows for a trade-off between training duration and performance.
>
> **Q:  Lack of quantitative evaluation and ablations for design choices.**
>
> Thank you for the suggestion! Quantitative evaluation remains challenging for 3D generation tasks. Furthermore, existing metrics primarily target full-shape appearance rather than part-level geometry. Given this limitation, we mainly rely on qualitative ablation studies to demonstrate design impacts visually, especially the noticeable improvements after fine-tuning the VAE.
>
> **Q:  Inconsistent part divisions and limited controllability.**
>
> We acknowledge this major limitation, illustrated in Figure 3 of the supplementary materials. This inconsistency largely arises from mixed-quality data, where objects within the same category often exhibit varying part divisions due to different workflows or styles of the artists. We have discussed potential future directions, such as incorporating 2D segmentation masks (e.g., from SAM) to guide more consistent and controllable 3D part division.
>
> **Q:  Lack of clarification on downstream tasks such as joint articulation and editing.**
>
> Thank you for highlighting this issue! Downstream tasks like articulation and editing are indeed primary motivations for our method. However, our current model, although promising, still exhibits robustness limitations, as evidenced by failure cases in Figure 3 of the supplementary materials. These limitations stem from mixed-quality datasets and the two-volume part-packing approach. We will clarify this limitation explicitly in the revised manuscript to prevent any potential overclaims.
>
> **Q:  How to generate unseen or occluded parts of the object?**
>
> Our model follows recent advances in 3D native diffusion models (e.g., CLAY, TripoSG, Hunyuan3D, TRELLIS), which first learn compact 3D latent representations from complete 3D shapes, and then train a diffusion/flow-matching model to map input image features to the latent space. Therefore, the generated 3D shapes are mostly complete. This solves some occlusion problems, for example, even if one chair's leg is unseen in the image, we can correctly predict all four legs based on the semantic clues. Our part-based model further allows for predicting each leg separately. However, the model may not be able to predict totally out-of-domain shapes due to the data-driven nature.

---

> ### Comment · Reviewer_Gc4w · 2025-08-02
>
> The author addressed my concerns well, so I will raise my judgment.
>
> Q: The method requires more computational resources.
> Thank you for the clarification, and after your sincere explanation, I understand your point.
>
> Q: Lack of quantitative evaluation and ablations for design choices.
> I see the limitation, and thank you for the clarification.
>
> Q: Inconsistent part divisions and limited controllability.
> This would be solved by collecting more data in the future, so it may not be a critical issue.
>
> Q: Lack of clarification on downstream tasks such as joint articulation and editing.
> Hope that you can update this on the final version.
>
> Q: How to generate unseen or occluded parts of the object?
> I understand better because of your clear explanation, and also please update in the final version.

---

### Official Review · Reviewer_42Py · 2025-07-01

**Clarity:** 4
**Significance:** 3
**Originality:** 4
**Rating:** 5
**Confidence:** 4

**Summary:**

This paper proposes a novel end-to-end framework for generating high-quality 3D objects with an arbitrary number of semantically meaningful parts from a single input image. The key innovation lies in the dual volume packing strategy, which allows the parallel generation of high-quality 3D objects with an arbitrary number of complete and semantically meaningful parts, improving efficiency and robustness. Sepecificaly, the authors introduce a heuristic algorithm to transform part connectivity graphs into bipartite graphs, enabling efficient packing of parts into two complementary volumes. This method demonstrates superior performance in terms of part-level and object-level generation.

**Questions:**

- **Choice of Volume Representation**: The paper utilizes a volume-based representation for 3D object generation, which, while effective, is known for its high computational and storage costs, especially as resolution increases. Given these limitations, have the authors considered alternative representations, such as implicit representations like NeRF or Gaussian fields? If so, what were the reasons for ultimately choosing the volume-based approach, and are there any plans to explore these alternatives in future work?

- **Evaluation Metrics**:The current evaluation relies heavily on cosine similarity-based metrics (ULIP, Uni3D), which primarily assess the alignment between generated meshes and reference images. While these metrics are useful, they do not provide a comprehensive evaluation of 3D generation quality. Have the authors considered incorporating additional metrics commonly used in 3D generation, such as GPTEval3D, user studies, or 3DGen-Score? These metrics could offer a more holistic view of the method's performance, including aspects like geometry plausibility, geometry details.

- **Downstream Application Details**:The teaser image showcases visualizations of downstream applications, but the paper lacks a detailed description of these experiments. Could the authors provide more information on the setup and results of these downstream tasks?  And what benefits did the proposed method bring compared to existing solutions?

**Ethical Concerns:**

["NO or VERY MINOR ethics concerns only"]

**Final Justification:**

My concerns have been addressed. I will keep my score as Accept.

**Limitations:**

yes

**Quality:**

4

**Strengths And Weaknesses:**

**Strengths**

- **Quality**： The paper is technically sound. The dual volume packing strategy effectively addresses the challenge of handling overlapping parts in 3D objects.  The results demonstrate superior performance compared to existing methods.
- **Clarity**：The paper is well-organized and clearly written. The authors provide detailed descriptions of their approach and experimental results.
- **Originality and Significance**： The dual volume packing strategy is a novel contribution. And the ability to generate part-level 3D meshes directly from images without relying on segmentation priors is a significant advancement.

**Weaknesses**

- **Evaluation Metrics**: The quantitative analysis relies solely on the cosine similarity between 3D and 2D features. While this metric effectively assesses the similarity between the generated meshes and reference images, it does not fully capture the quality of the 3D generation, such as shape fidelity, part completeness, or structural consistency.  A more comprehensive evaluation is expected.

- **Downstream Applications**:The paper provides limited discussion on the downstream applications of the proposed method. While the teaser posts potential applications in *Joint Articulation*  and *Editing*, there is no detailed exploration of how the method could be integrated into these fields or what specific benefits it would bring.

---

> ### Author Rebuttal · Authors · 2025-07-28
>
> Thank you for your valuable time and insightful comments! We have tried to address your concerns in our rebuttal:
>
> **Q:  Lack of evaluation on shape fidelity, part completeness or structural consistency.**
>
> Quantitative evaluation of 3D generation, especially at the part-level, remains very challenging. Current rendering-based benchmarks (GPTEval3D, 3DGen-Bench) are optimized for appearance rather than geometry, which is not the focus of our method. Also, due to limited part-wise evaluation methods and the complexity of user studies in this context, we emphasize qualitative evaluation. To encourage future evaluation advancements, we will release our code and a demo to facilitate comparisons.
>
> **Q:  Limited discussion on the downstream applications such as joint articulation and editing.**
>
> Thank you for highlighting this issue! Downstream tasks like articulation and editing are indeed primary motivations for our method. However, our current model, although promising, still exhibits robustness limitations, as evidenced by failure cases in Figure 3 of the supplementary materials. These limitations stem from mixed-quality datasets and the two-volume part-packing approach. We will clarify this limitation explicitly in the revised manuscript to prevent any potential overclaims.
>
> **Q:  Choice of volume generation compared with alternatives such as implicit NeRF or GS.**
>
> We adopted volume-based latent diffusion mainly following recent advancements (e.g., Shape2Vecset, CLAY, TripoSG). Regarding computational cost, since the diffusion model is trained on the compact latent space, it's not significantly different from other representations, and avoids rendering-based limitations (e.g., occlusion) of NeRF and GS. Nevertheless, our core part-packing concept is representation-agnostic, allowing flexibility in adopting alternative representations if advantageous.

---

### Official Review · Reviewer_7qd5 · 2025-07-03

**Clarity:** 3
**Significance:** 3
**Originality:** 3
**Rating:** 4
**Confidence:** 5

**Summary:**

This paper proposes a new 3D part generation method based on Dual Volume Packing. The authors first extract a bipartite graph from the 3D mesh to encode dual volume latents. A flow model is then employed to generate these latents. The authors compare their proposed method with state-of-the-art 3D generators.

**Questions:**

See the weaknesses.

**Ethical Concerns:**

["NO or VERY MINOR ethics concerns only"]

**Final Justification:**

My concerns are addressed. I tend to keep my score.

**Limitations:**

Yes

**Quality:**

3

**Strengths And Weaknesses:**

Strengths:

1. The proposed Dual Volume Packing approach is novel and reasonable. The dual volume latents can support a large number of parts, as all parts are divided into two volumes.

2. The proposed method demonstrates comparable global results with state-of-the-art 3D generators while achieving effective part decomposition.

3. The presentation is clear and easy to follow.

Weaknesses:

1. To extract all meshes in the dataset into a bipartite graph, the authors propose a sequence of edge contraction operations. However, these operations require merging parts, which may compromise the original part information in the mesh.

2. The authors do not provide an analysis of part consistency, as all parts within one volume are disconnected. I have concerns about the consistency of combinations, particularly regarding mesh penetration issues when parts from the two volumes are merged.

3. Minor issue: the current framework does not generate texture.

---

> ### Author Rebuttal · Authors · 2025-07-28
>
> Thank you for your valuable time and insightful comments! We have tried to address your concerns in our rebuttal:
>
> **Q:  The part-merging in data preprocessing may compromise the original part information.**
>
> We strive to preserve original part annotations in our data pipeline and only merge parts when absolutely necessary. Specifically, when GLB meshes provide meta-information regarding geometry groups, we directly use these annotations. Otherwise, we fallback to treating each connected component as an individual part. However, we observed many issues caused by operations like UV-unwrapping, where surfaces of the same semantic part are split for texturing ease. Therefore, empirical part-merging rules are applied to recombine these disconnected surfaces appropriately.
>
> **Q:  Lack of analysis on part consistency when parts from two volumes are merged.**
>
> We consider minor overlaps or penetrations between parts acceptable, as they frequently appear in artist-created meshes. Our approach effectively reproduces such behavior, as demonstrated in Figure 5 and supplementary videos. Even when overlaps occur, each generated part remains individually complete and watertight, since they are generated independently in separate volumes.
>
> **Q:  No support of texture generation.**
>
> Thank you for your suggestion! Similar to other 3D-native diffusion models (e.g., CLAY, TripoSG, Hi3DGen), we currently focus on improving geometry generation quality. Texture model typically leverages a distinct 2D multi-view diffusion pipeline, differing significantly from the geometry model. Therefore, we consider texture generation outside the immediate scope of this work.

---

> > ### Comment · Reviewer_7qd5 · 2025-08-03
> >
> > Thanks for the rebuttal. My concerns are addressed.

---

### Official Review · Reviewer_5MKr · 2025-07-03

**Clarity:** 2
**Significance:** 3
**Originality:** 3
**Rating:** 4
**Confidence:** 3

**Summary:**

The paper presents an end-to-end framework for generating part-level 3D objects from single images using a dual volume packing strategy. The key idea is converting part-connectivity graphs into bipartite graphs, allowing all parts to be packed into two volumes where non-contacting parts can be naturally separated. The method extends the existing 3D generation model to simultaneously generate two latent codes that decode to two volumes. The experiments shows visually better segmentation compared to prior methods.

**Questions:**

Please see weakness above.

While I appreciate the technical novelty of the dual volume packing strategy for addressing efficiency in part-level 3D generation, I have concerns about the method's broader impact and extensibility. The efficiency gains, though good, may not address the most critical challenges in this domain. More pressing issues include faithfull reconstruction of input details, consistent part generation, and geometric constraints between parts - areas where this method shows limited promise for improvement.

**Ethical Concerns:**

["NO or VERY MINOR ethics concerns only"]

**Final Justification:**

The authors addressed my concerns. I tend to keep my current score and lean toward accepting the paper.

**Limitations:**

yes

**Quality:**

3

**Strengths And Weaknesses:**

Strengths:

1. The dual volume packing strategy that converts 3D shapes into two separate volumes containing disjoint parts is well-motivated and enables efficient generation. The approach is solid and produces convincing results.
2. The dataset processing pipeline with part extraction, repair and filtering is important for the community. Open-sourcing this pipeline would benefit future research in part-level 3D generation.
3. The presented results demonstrate visually appealing part-level 3D generation with reasonable separation and overall shape.

Weakness:

1. The part divisions generated depend entirely on the model's learned representations from training data, and the presented representation has no mechganism for consistency guarantee. For example, in the Figure 4 row 2 example, the method fails to faithfully generate the four eyelashes visible in the input image. This limitation cannot be easily addressed by the segmentation mask approach mentioned in the authors' limitation discussion.
2.  The method lacks mechanisms to enforce geometric and topological constraints between parts. The required post-processing steps further reduce user control over the final output and limit the method's applicability for precision applications.
3. The paper would benefit from additional examples in the image-to-3D comparison beyond those currently presented. More diverse test cases would better demonstrate the method's capabilities and limitations.
4. On the comparison experiment, we still don't know how much the performance difference is because of the training data, since there is no experiment training the prior method on the new data or vice-versa.

Minor Weakness:

5. While not strictly necessary, developing metrics specifically for part-level 3D generation quality, consistency, and semantic correctness would strengthen the evaluation. Current comparisons rely on general generation quality metrics, and part-specific metrics would benefit future research and enable more meaningful comparisons in this domain.

---

> ### Author Rebuttal · Authors · 2025-07-28
>
> Thank you for your valuable time and insightful comments! We have tried to address your concerns in our rebuttal:
>
> **Q: The part divisions generated depend entirely on the model's learned representations from training data, and the presented representation has no mechanism for consistency guarantee.**
>
> We acknowledge that our data-driven part division approach has limitations, especially when the dataset is of mixed quality. As illustrated by the Sponge Bob example, even whole-shape baselines struggle to generate fine details such as the four eyelashes. Given that our part-packing strategy is model-agnostic, we anticipate improvements in part consistency and quality with better backbones and higher-quality datasets.
>
> **Q:  Lack of mechanisms to enforce geometric and topological constraints between parts.**
>
> Similar to the previous point, our model primarily relies on dataset-provided part information to learn accurate geometric and topological relationships. Although the model uses attention layers to exchange information between different parts implicitly, the generated parts may still be inconsistent or suboptimal with certain random seeds.
>
> **Q:  More test cases and comparisons to better demonstrate the applicability of the model.**
>
> Thank you for this suggestion! We will create a comprehensive project page featuring additional samples and provide open-source code with a demo to allow users thorough testing of our model.
>
> **Q:  Influence of training data on the comparisons.**
>
> Due to our part-related filtering and preprocessing steps, our dataset contains approximately 254K samples, smaller than TripoSG (2M samples including private datasets) or Hi3DGen (870K including synthetic samples). However, since most baselines majorly rely on subsets of Objaverse and Objaverse-XL for training, it's still relatively fair to compare with.
>
> **Q:  Metrics for part-level 3D generation quality, consistency and semantic correctness.**
>
> Thank you for this suggestion! Currently, no standard metrics exist for evaluating part-level generation quality in 3D generation. Developing robust, automated metrics in this domain alone remains a difficult research challenge, which is why we still rely predominantly on qualitative assessments and will note this limitation in the revised manuscript.

---

> > ### Comment · Reviewer_5MKr · 2025-08-05
> >
> > Thank you for your explanation. I'm leaning toward accepting the paper.

---

### Decision · Program_Chairs · 2025-09-17

**Decision:**

Accept (poster)

**Comment:**

This paper proposes a method for part-based 3D shape generation from a single image. It has received all positive reviews.

The paper was found well organized, clear, easy to follow. The method was found well motivated, novel, solid, technically sound, efficient, important for the community. The results were found convincing, comparable or superior to state-of-the-art, visually appealing,

Concerns included the lack of consistency guarantee originating from the segmentation mask approach and the lack of relevant analysis, the requirement to merge parts, the lack of mechanisms to enforce geometric and topological constraints between parts, the insufficient examples, the insufficient metrics, the lack of ablation on training data, the limited discussion, the high training cost, the dependence on scale, the inclusion of visual examples on functionalities without empirical support, the lack of a user study.

The authors provided detailed feedback in their rebuttal. This feedback addressed all reviewers' concerns or it was acknowledged that there are limitations to be discussed in the paper, which is still okay. The authors committed to update the paper according to the new material they have presented during the discussion.

It is therefore recommended to accept the paper.